# High-Temperature Disaster Risk Assessment for Urban Communities: A Case Study in Wuhan, China

**DOI:** 10.3390/ijerph19010183

**Published:** 2021-12-24

**Authors:** Zhuoran Shan, Yuehui An, L’ei Xu, Man Yuan

**Affiliations:** 1School of Architecture and Urban Planning, Huazhong University of Science and Technology, Wuhan 430074, China; hust_szr@sina.com (Z.S.); ayh_hya@163.com (Y.A.); 2China Nerin Engineering Company Limited, Nanchang 330199, China; xulei@nerin.com

**Keywords:** high-temperature disaster, risk assessment, built environment, GWR model, Wuhan, community

## Abstract

High-temperature risk disaster, a common meteorological disaster, seriously affects people’s productivity, life, and health. However, insufficient attention has been paid to this disaster in urban communities. To assess the risk of high-temperature disasters, this study, using remote sensing data and geographic information data, analyzes 973 communities in downtown Wuhan with the geography-weighted regression method. First, the study evaluates the distribution characteristics of high temperatures in communities and explores the spatial differences of risks. Second, a metrics and weight system is constructed, from which the main factors are determined. Third, a risk assessment model of high-temperature disasters is established from disaster-causing danger, disaster-generating sensitivity, and disaster-bearing vulnerability. The results show that: (a) the significance of the impact of the built environment on high-temperature disasters is obviously different from its coefficient space differentiation; (b) the risk in the old city is high, whereas that in the area around the river is low; and (c) different risk areas should design built environment optimization strategies aimed specifically at the area. The significance of this study is that it develops a high-temperature disaster assessment framework for risk identification, impact differentiation, and difference optimization, and provides theoretical support for urban high-temperature disaster prevention and mitigation.

## 1. Introduction

In recent decades, the trend of global warming has become obvious, and the average temperature continues to rise. The frequency of extreme high-temperature events has increased worldwide, and the duration of urban high-temperature heatwave events has become longer under the dual effects of global warming and rapid urbanization. High temperatures have gradually become a serious meteorological disaster and a hot issue in climate change research [1,2]. Urban high-temperature disasters not only consume a large amount of energy but also cause great threats to the physical and mental health of urban residents [3]. High-temperature disasters may increase the risk of heatstroke, respiratory and cardiovascular diseases, and even mortality [4,5,6,7]. The annual average death toll caused by the negative impact of high-temperature heatwaves is much higher than that of other extreme weather events [8]. In 2003, the European summer heatwave claimed 35,000 lives [9]. In the summer of 2021, Sacramento, Phoenix, and other cities in the United States reached the “level 4” high-temperature risk level. Nearly 200 people died in the states of Oregon and Washington due to a heatwave (BBC), while Las Vegas in Nevada recorded its highest temperature (NWS) since 1942. In 2013, China experienced a rare high-temperature heatwave. As the hardest hit area, Shanghai recorded 1347 heat-related deaths [10]. Because of the frequent occurrence of extreme high-temperature events and the great threat to people’s lives and health, many scholars have conducted research on the risk assessment of high-temperature disasters in recent years [8,11,12].

It is generally believed that the built environment is an important factor affecting urban high-temperature disasters, and its direct manifestation is the urban heat island phenomenon [13]. Urban development intensity (e.g., floor area ratio, building density) is the main driver of urban heat islands and can change the diffusion of heat in a city [14,15,16]. Increased building density raises land surface temperature (LST), while an increase in building height weakens LST [17,18]. Weng et al. (2007) determined the effects of different types of land cover, such as impervious land (artificial surfaces), bare land, vegetation, and water area, on surface temperature, and confirmed that LST was positively correlated with impervious surface fraction, but negatively correlated with green vegetation fraction [19]. Although such factors as land cover and development intensity have been proven to have a significant impact on the urban thermal environment, descriptions of the built environment in existing research are mostly based on a macro scale, and the factors of the built environment in different areas of a city are not optimized [20]. By contrast, focusing on urban basic units, research on the influence of the built environment on the regional differentiation of the urban thermal environment on a community scale is still in the exploratory stage.

At present, quantitative methods for determining the index weight of urban high-temperature disasters have been studied to some extent. Aubrecht and Ozceylan (2013) normalized the impact indicators and analyzed high-temperature vulnerability disasters in Washington using equal-weight superposition [21]. Reid et al. (2009) used principal component analysis to analyze the impact of the natural environment, social economy, and other factors on high-temperature disasters in the United States and mapped community determinants of such disasters [22]. Ying and Fang (2020) used standard deviation ellipse and spatial principal component analysis to explore the thermal environment effect of human settlements in Changsha and identified the relationship between natural and cultural factors and the urban thermal environment effect [23]. Zhu et al. (2014) used an analytic hierarchy process and principal component analysis to determine the weight of indicators and explore the spatial distribution differences of high-temperature disasters in Guangdong Province [24]. It is easy to reveal the spatial pattern of high-temperature disasters using the graph stacking method, but this method makes it difficult to reflect the main influencing factors of high-temperature disaster risk. Principal component analysis can be used to determine the weight graph overlay method of indicators, but the results depend on the selection of indicators to a great extent [25]. By contrast, the geographically weighted regression model focuses on the differences and variability of regression coefficients in geographical space [26]. The model incorporates spatial position information into the regression equation, which can reflect the spatial non-stationarity of the parameters, and the index weight can be more in line with objective reality [27]. Compared with the ordinary least squares (OLS) model, the geography-weighted regression (GWR) model has the following advantages: it explores the spatial law through parameter estimation of the model by GIS; it better reflects the local characteristics of the relationship between variables, obtaining smaller residual errors; and it yields a more significant statistical test of the model [28].

Research on the comprehensive assessment framework of high-temperature disaster risk is more mature than research on the factor selection and indicator weighting of high-temperature disasters. Ma (2012) comprehensively classified the high-temperature disaster risk areas in Luoyang based on four dimensions: disaster-causing factors, disaster-generating environment, disaster-bearing bodies, and disaster prevention and mitigation [29]. Shui et al. (2017) constructed a spatial evaluation index system of high-temperature vulnerability in Fuzhou with coupling adaptability from three aspects: urban high-temperature exposure, vulnerability, and adaptability [11]. At present, the commonly used vulnerability assessment framework is the exposure–sensitivity–adaptability framework proposed by the Intergovernmental Panel on Climate Change (IPCC) in its third assessment report [8].

Overall, there are three restrictions in the risk assessment of high-temperature disasters. (1) The spatial heterogeneity of the community construction environment is often ignored. High-temperature disasters are not uniformly distributed in all regions. In many studies, it is difficult to identify areas with high-temperature disaster risk and reveal the differences in spatial patterns of high-temperature disaster risk, because local spatial changes and the relationship between influencing factors are not taken into account. Therefore, the planning significance of the high-risk areas is weak [11]. (2) Risk assessment of high-temperature disasters from the community unit level is almost lacking. In the urban center of China, there are large areas of old communities with poor quality and unsafe environments due to population aggregation, traffic congestion, and outdated facilities [30]. The urban community, as an important space in people’s daily lives, can prevent high-temperature disasters directly related to residents’ physical and mental health [31]. However, the current research mainly focuses on the macro and medium levels, such as the country or city, and lacks the risk assessment of high-temperature disasters at the community unit level. (3) The effects of population distribution are rarely considered. As people in urban communities are the main group affected by heatwaves, research on population distribution cannot be ignored [32]. Age is an important factor affecting vulnerability to high temperatures, and the elderly are more susceptible to heat exposure [21,33]. At present, few studies on urban heat disaster risk consider population distribution, especially the exposure risk of vulnerable groups.

This study assesses the risk of high-temperature disasters in urban communities in downtown Wuhan, China. First, GWR is used to explore the spatial differences of influencing factors. Second, a risk assessment model is established to evaluate high-temperature disasters from the perspective of disaster-causing danger, disaster-generating sensitivity, and disaster-bearing vulnerability. Finally, planning strategies are proposed to reduce high-temperature disasters and promote the resilience of high-risk communities.

## 2. Data and Methodology

### 2.1. Study Area and Data

Wuhan is a typical “furnace” city in the middle and lower reaches of the Yangtze River, and has been seriously affected by high-temperature disasters. Wuhan is located at 29°58′–31°22′ N, 113°41′–115°05′ E. It is the capital city of Hubei Province and the only megacity in central China [34]. Downtown Wuhan, which is composed of seven central districts (Zhuchengqu), is the study area, covering a total area of 432 km^2^ (Figure 1). It is located in the plain area of the middle and lower reaches of the Yangtze River, with an altitude between 19 and 20 m. Downtown Wuhan has a subtropical humid monsoon climate with abundant perennial rainfall, sufficient heat, and four distinct seasons. The annual average temperature is 15.8–17.5 °C, and the most extreme maximum temperature recorded is 41.3 °C. The annual total sunshine hours are between 1810 and 2100 h, and annual precipitation is between 1150 and 1450 millimeters. Wuhan is historically known as the City of a Hundred Lakes, as hills, rivers, and lakes form the city landscape. Downtown Wuhan is divided into three parts (Hankou, Wuchang, and Hanyang) by the Yangtze River and the Han River; the largest three lakes are East Lake (32.8 km^2^), South Lake (7.6 km^2^), and Sha Lake (3.3 km^2^). Due to data availability, 973 communities are analyzed in the study, and the average, minimum, and maximum area values are 0.43 km^2^, 0.01 km^2^, and 9.02 km^2^, respectively. All communities were classified into 18 residential groups. The residential population in the study area is 6.7 million, accounting for 60.28% of the city’s total population. The population of the elderly over 65 years of age and children under six years accounted for 11.98% and 2.51%, respectively.

The temperature of downtown Wuhan is usually 1–2 °C higher than that of other regions, including the suburbs. In recent years, 2017 recorded the highest air temperature in Wuhan, 39.7 °C, and there were 21 days with temperatures above 35 °C. The highest land surface temperature was 57.1 °C, and there were 111 days with land surface temperatures above 35 °C.

Remote sensing data from Landsatx 8 on 27 August 2017 were used to retrieve land surface temperature in downtown Wuhan. A spatial resolution of 30 m of the data is suitable for assessment in communities. According to the meteorological monitoring data, the weather was fine on 27 August 2017, with a sunshine duration of 10.7 h. The highest surface temperature was 38.5 °C, the lowest surface temperature was 26.7 °C, and the average wind speed was low. After correcting radiometric and geometrical distortion, the radiative transfer equation-based method was used to convert the digital numbers of Landsat
8 to at-sensor brightness temperatures [35], and the land surface temperature of each pixel was retrieved. The community LST was calculated by averaging all values of pixels in the extent, and pixels on building roofs were excluded [36]. GIS data include land cover maps, building data, road data, and demographic data of the communities.

### 2.2. Methodology

The study developed a framework for high-temperature risk assessment for urban communities, with risk consisting of three dimensions: disaster-causing danger, disaster-generating sensitivity, and disaster-bearing vulnerability.

To build the metric and weight system of risk assessment, the effects of built environment factors on community LST were first explored. According to past studies, built environment factors include land cover (impervious land coverage, bare land coverage, vegetation coverage, and proximity to water) and urban development intensity (floor area ratio and building density) [37,38]. The percentage of impervious land and bare land in each community was calculated as impervious land coverage and bare land coverage, and the average value of the normalized difference vegetation index (NDVI) in each community was used to measure the vegetation coverage. The NDVI was calculated using Band 3 (red) and Band 4 (near-infrared) of Landsat 8, and the range of NDVI was between −1 and 1. A higher NDVI indicates a higher level of vegetation coverage. The proximity to water was calculated by overlaying the buffering areas of rivers and lakes on communities, with the buffering radius based on the size of the water area [39]. The larger the water area, the greater the cooling distance range. The closer to water, the better the cooling effect. Based on GIS building data, the metrics of floor area ratio (FAR) and building density were calculated for each community. The community LST and built environment metrics were used as the dependent variable and independent variables, respectively, in an OLS regression, and insignificant variables were removed from the metric system of risk assessment. Variance inflation factor (VIF) is a tool to help identify the degree of multicollinearity. Mathematically, the VIF for a regression model variable is equal to the ratio of the overall model variance to the variance of a model that includes only that single independent variable. A high VIF indicates that the associated independent variable is highly collinear with the other variables in the model [40]. A multicollinearity test was carried out to eliminate the metrics with tolerance less than 0.1 or variance inflation factor greater than five. To consider spatial non-stationarity or geographical heterogeneity, a GWR model was used to analyze the spatially varying relationships between the dependent and independent variables using the software GWR4.0. The GWR model can be a powerful tool to explore the spatially different effects of built environment metrics on LST across communities in downtown Wuhan, through which local specific weights of metrics for high-temperature risk assessment can be developed. Moran’s I value is used to describe the average correlation degree of all spatial units with surrounding areas in the whole region. If Moran’s I value is positive, it indicates a clustering trend; otherwise, if Moran’s I value is negative, it indicates a discrete trend [41]. All data were normalized before employing the OLS and GWR models. Figure 2 shows the spatial patterns of influencing variables and Table 1 presents the statistics of built environment factors.

The level of high-temperature disaster risk in urban communities was evaluated using three maps of disaster-causing danger, disaster-generating sensitivity, and disaster-bearing vulnerability. Disaster-causing danger, or the driving factors of high-temperature disasters in communities, was measured using metrics of development intensity, bare land coverage, and impervious land coverage. Disaster-generating sensitivity is associated with natural environmental factors related to high-temperature disasters, and metrics of vegetation coverage and proximity to water were used to evaluate disaster-generating sensitivity. The higher the value of vegetation coverage and proximity to water in communities, the lower the level of disaster-generating sensitivity. The local B coefficients in the GWR results were used as the local weights of the metrics in estimating disaster-causing danger and disaster-generating sensitivity. Disaster-bearing vulnerability refers to the vulnerability of the population exposed to high-temperature disasters, and the weighted sum value of population densities of different ages in each community was calculated for this metric: children (under 18 years old), young people (19–39 years old), middle-aged people (40–64 years old), and the elderly (over 65 years old). For the sake of comparison, the values of disaster-causing danger, disaster-generating sensitivity, and disaster-bearing vulnerability were normalized across all the communities, and a comprehensive risk assessment of high-temperature disaster was conducted by integrating the three maps.

## 3. Results

This section presents the following results: (a) description of the spatial heterogeneity of temperature distribution; (b) identification of the influencing factors in the weight system; and (c) analysis of the risk of high-temperature disaster in urban communities from three dimensions.

### 3.1. Description of Temperature Distribution

Figure 3 shows the map of community LST in downtown Wuhan. The average, maximum, and minimum values of community LST were 39.4 °C, 45.6 °C, and 32.9 °C, respectively. The community LST was concentrated in the range of 37.9–39.9 °C, accounting for 45.37% of the total area. The natural breakpoint method was used to classify communities into five grades: high LST area (42.0–45.6 °C), relatively high LST area (40.3–42.0 °C), medium LST area (38.8–40.3 °C), relatively low LST area (37.1–38.8 °C), and low LST area (32.9–37.1 °C). There were 277 (28%) communities in relatively high LST areas and 322 (32%) communities in medium LST areas, with downtown Wuhan showing a high level of LST. There was a spatial imbalance in LST between Hankou, Hanyang, and Wuchang. In general, the average LST in Hankou was the highest (40.1 °C), and the value of Wuchang was the lowest (38.7 °C). In the 1st Ring Road, most of the high LST areas were concentrated in old communities in central Hankou and central Wuchang. In the 2nd Ring Road, the spatial agglomeration of high LST became weak, and relatively high LST areas were located along the main roads. The distribution of medium LST communities was scattered, and these areas were mainly located around relatively high LST communities. Low LST communities were spatially concentrated, with most of them mainly distributed along the river and around water, such as East, South, Moshui, and Jinyin lakes, and the south bank of the Han River. Global spatial autocorrelation was used to analyze the spatial pattern of community LST. The global Moran’s I value was 0.45, showing a significantly positive spatial autocorrelation, and the community LST was not spatially independent. The GWR model performed better than the OLS model when dealing with the problem of spatial autocorrelation.

### 3.2. Construction of Weight System

Table 2 presents the results of the OLS regression and multicollinearity tests. Bare land coverage failed to pass the significance test, and impervious land coverage was removed because of multicollinearity. Hence, the metrics of risk assessment include NDVI, proximity to water, FAR, and building density, and they were used as independent variables in the GWR model. The statistics of the standardized B coefficients are shown in Table 3. The DIFF criterion was used to test the local non-stationarity of each independent variable, where a DIFF value less than 0 indicates the spatial stationarity of the variable. The results show that there was spatial heterogeneity for all metrics. The GWR model had a higher R^2^ and lower AICc than the OLS model, suggesting that the GWR model had better overall performance. According to the average value of B coefficients, building density had the greatest influence on community LST, followed by proximity to water and NDVI, while the effect of FAR was small. The spatial patterns of the local B coefficients are shown in Figure 4. For most communities, NDVI (90%), proximity to water (98%), and FAR (99%) had negative effects on community LST, while building density (100%) had a positive effect on community LST.

The increase in NDVI was conducive to the mitigation of community LST, and the negative effect was higher in areas outside the 2nd Ring Road, such as Huangpu, Houhu, Gutian, Shisheng, and Baishazhou. Proximity to water decreased the community LST, and the negative effect was higher in Huangpu, Gutian, Shisheng, Wugang, and central Hankou. The effect of building density on community LST was positive in all communities. Most buildings are built with high thermal conductivity materials, such as cement and reinforced concrete, through which mass heat passes from solar heat radiation to the ground in communities. At the same time, most of the land is occupied by high-density buildings, and it is difficult for heat to evaporate in a timely manner. After controlling for building density, FAR had a significant negative effect on community LST. High-rise buildings can block sunlight and form a large, shaded area, thereby reducing community LST. These local B coefficients in each community were used as the weights of the metrics in risk assessment. FAR and building density were weighted-summed to measure disaster-causing danger, and NDVI and proximity to water were a weighted-summed measure of disaster-generating sensitivity.

### 3.3. Assessment of Disaster Risk

#### 3.3.1. Disaster-Causing Danger

The average, maximum, and minimum values of disaster-causing danger in downtown Wuhan were 0.46, 1, and 0.02, respectively. The natural breakpoint method was used to classify communities into five grades: high danger area (0.78–1), relatively high danger area (0.56–0.77), medium danger area (0.37–0.55), relatively low danger area (0.17–0.36), and low danger area (0–0.16). In total, there were 361 communities (37%) located in (relatively) high danger areas, and downtown Wuhan showed a high level of disaster-causing danger. Most high danger communities were located in central Hankou and central Wuchang in the 1st Ring Road, which has many old communities built before 2000 with high-density multi-story buildings. High development intensity reduced the possibility of convective heat dissipation and increased disaster-causing danger. Taking central Wuchang as an example, many old communities are located in an ancient city historical reserve area, where high-rise buildings are not allowed to replace high-density, low-rise buildings. Additionally, some large commercial centers, such as Hanzheng Street, Jianghan Road, and Simenkou, are located in this region, which draws a large number of people and traffic with intensive artificial heat sources. In the region of the 2nd Ring Road and the 3rd Ring Road, disaster-causing danger becomes weak. After the 1998 reform of the housing system and the 2003 land bid invitation, auction, and listing system, many new residential communities were built in this area. Although it accommodates a large population, the building density in newly built communities is much lower than that in the 1st Ring Road. Some (relatively) high danger communities are located near the 2nd Ring Road, and there are many urban villages in this area. Compared with central urban areas, these urban villages provide residential space for a large number of rural workers, and the environmental quality is quite poor with overly high building density. Furthermore, there are some urban industrial parks in this region, and industrial production increases the disaster-causing danger of nearby communities. High-intensity urban development near large lakes (e.g., East, South, Moshui, and Sha Lake) is restricted, and the levels of disaster-causing danger are relatively low.

#### 3.3.2. Disaster-Generating Sensitivity

The average, maximum, and minimum values of the disaster-generating sensitivity were 0.29, 1, and 0.02, respectively. The natural breakpoint method was used to classify communities into five grades: high sensitivity area (0.56–1), relatively high sensitivity area (0.30–0.55), medium sensitivity area (0.19–0.29), relatively low sensitivity area (0.09–0.18), and low sensitivity area (0–0.08). There were 334 communities (34%) in relatively high sensitivity areas and 445 communities (45%) in relatively low sensitivity areas. In general, the disaster-generating sensitivity in downtown Wuhan was low, and water area and green land helped to reduce disaster-generating sensitivity. The overall vegetation coverage in downtown Wuhan was medium, and the average, maximum, and minimum values of NDVI were 0.46, 0.81, and 0.24, respectively. The majority of communities (410, 42%) had an NDVI between 0.39 and 0.49, and only 125 (13%) communities had an NDVI larger than 0.6. The NDVI in old communities in central Hankou and central Wuchang was low, and the NDVI of communities along the 3rd Ring Road was high. As the global and local B coefficients show, water area had a larger effect on community LST than vegetation had. Water resources are rich in downtown Wuhan; the Yangtze River, the Han River, and large lakes (East Lake, South Lake, and Sha Lake) form large open spaces and air vents. Low sensitivity and relatively low sensitivity communities were mainly distributed around these rivers and lakes. For example, the vegetation coverage in old communities in central Hankou along the Yangtze River was low, but a high level of proximity to water decreased the disaster-generating sensitivity in this area. High sensitivity communities were spatially aggregated in the northwest of central Hankou, Tazihu, Gutian, Nanhu, and Guanshan, and these areas were far away from rivers and lakes. In Wuhan City General Planning, six large wedge-shaped ecological spaces from suburban to downtown Wuhan in six directions are planned in the future, and the high level of disaster-generating sensitivity in the northwest, south, and southeast directions suggests the need for planning.

#### 3.3.3. Disaster-Bearing Vulnerability

Compared with young and middle-aged people, children and the elderly are vulnerable groups, and their health may face greater threats in front of high-temperature disasters. In a society with low fertility and increasing aging at this time, it is necessary to pay more attention to the health risks of the elderly and children. Higher weights (3) were given to the population density of children and elderly, and lower weights (2) were given to the population density of young and middle-aged people. The average, maximum, and minimum values of the disaster-bearing vulnerability were 0.35, 1, and 0.02, respectively. The natural breakpoint method was used to classify communities into five grades: high vulnerability area (0.77–1), relatively high vulnerability area (0.54–0.76), medium vulnerability area (0.33–0.53), relatively low vulnerability area (0.14–0.32), and low vulnerability area (0–0.13). There were 280 communities (28%) in relatively high vulnerability areas and 533 communities (54%) in areas with relatively low vulnerability. The city center on the 1st Ring Road had the largest population density, and the level of disaster-bearing vulnerability decreased outward from central Hankou and central Wuchang to the urban fringe. Old communities in the city center accommodated a large number of the elderly, and the highly vulnerable areas were mainly concentrated in this region. As urban sprawl has beset the city, many young people have settled with their children within the 2nd Ring Road, and most of the communities with medium and relatively high vulnerability were located there. Except for Tazihu, Qingshan, and Guanshan, the population density of children and the elderly was low outside the 2nd Ring Road, and areas with (relatively) low vulnerability were mainly located there.

#### 3.3.4. Comprehensive Risk

A comprehensive risk map was developed by spatially overlaying maps of disaster-causing danger, disaster-generating sensitivity, and disaster-bearing vulnerability; in addition, a pivot table was generated (Table 4). There is a spatial mismatch between the danger map and sensitivity map, and communities with both (relatively) high values of disaster-causing danger and disaster-generating sensitivity accounted for 16% of downtown Wuhan. For these communities, development intensity was quite high, and nearby ecological resources were poor. There were 387 (40%) communities with only relatively high values of disaster-causing danger or disaster-generating sensitivity, and the remaining 434 (45%) communities were in medium and low danger and sensitivity areas. Highly vulnerable communities were spatially consistent with high disaster-causing danger, as most of the communities with (relatively) high vulnerability (65%) were located in areas of (relatively) high danger. However, only 34% of the communities with relatively high vulnerability were located in areas with (relatively) high sensitivity. Hence, most vulnerable groups, such as the elderly and children, were highly exposed to high disaster-causing danger and low disaster-generating sensitivity. In general, there were 65 (7%) communities with the highest comprehensive risks (high levels of disaster-causing danger, disaster-generating sensitivity, and disaster-bearing vulnerability), and most of these areas were old communities in central Hankou and Wuchang or urban villages near the 2nd Ring Road. Low comprehensive risk communities were mainly located along the river or around large lakes (Figure 5).

## 4. Policy Implications

In China, rapid urbanization has consumed large land resources, and high-density urban development and insufficient green land may increase the risk of high-temperature disasters in the context of global warming [42,43]. At present, Wuhan is in an important transition period from a central city in central China to a national central city and an international metropolis. Both urban construction and rural construction have shifted to stable and orderly growth, and urban development has shifted from expansion to promotion, which is very likely to lead to the risk of high-temperature disasters. The policy implications are discussed from three aspects: public health, blue–green network and resource elements.

(1) Public health should be improved comprehensively. Bowler et al. (2010) have confirmed the potential of urban green infrastructure to mitigate high temperatures [44]. Green Infrastructure creates urban ecosystem settings within which the socioeconomic and other aspects of public health exist [45]. Highlighting disadvantaged groups that are especially vulnerable to the effects of high-temperature disasters is centrally important to communities as they develop policy implications [46]. In the community public space, shade and recreational places should be provided for children. At the same time, barrier-free design should be applied to improve the possibility of outdoor activities for the elderly. It is necessary to create a livable community and comfortable living environment to effectively promote the physical and psychological health of residents.

(2) The blue–green network should be created scientifically. The heat capacities of vegetation and water are obviously higher than those of artificial land surfaces, and a blue–green interwoven built environment could effectively reduce the heat gain rate under solar heat radiation through transpiration and evaporation [47]. Greenways could be used to connect large comprehensive lake and mountain parks to build a network of parks and green spaces, and greenways on main roads and non-motorized lanes could be improved to connect communities with parks, squares, schools, and places where residents go about their daily activities. For high sensitivity communities far away from rivers and lakes, it is necessary to strengthen the landscape pattern of interspersed blue and green for the entire city.

(3) The resource element should be managed reasonably. In different countries, the resource elements around nature protection and sustainable development have been widely used in the planning system [48]. At the legal level, all ecological elements of hills, forests, grasslands, lakes, and rivers should be included in the ecological protection red line for strict protection [31]. Wuhan is rich in natural resources of hills, rivers, and lakes, which helps to reduce disaster-generating sensitivity. Several high-quality green corridors could be built to connect rivers and lakes to develop a blue–green urban network by building the ecological axis of the Yangtze River and the Han River and six large wedge-shaped green lands. The environment of waterfront lines, ribbon corridors, and small and micro green spaces should be improved to form a comfortable, continuous, and diverse waterfront network. The network could serve as a bridge between citizens’ daily lives and waterfront culture to connect the community with rivers and lakes. With the control and integration of various resource elements, the resilience of the community can be effectively enhanced, thus reducing the risk of high-temperature disasters.

Most old communities face a high level of disaster-causing dangers, and environments are quite poor with high-density buildings, narrow alleys and streets, insufficient green land, and high population density. The *14th Five-Year Plan for National Economic and Social Development* proposed action on urban renewal and reconstruction of old urban communities. From 2019, China aims to reconstruct 170,000 old urban communities, affecting hundreds of millions of people in three years. According to the three-year plan for the *renovation of old communities in Wuhan (2019*–*2021)*, more than 200 old communities will be renovated in downtown Wuhan by 2021 [49]. To reduce the risk of high-temperature disasters, this study proposes several planning strategies for urban renewal in downtown Wuhan (Table 5).

(1) For communities being built or to be built, give priority to the organization of land use structure and the optimization of architectural composition. In terms of floor area ratio, a certain building height of 24 m or more could be maintained to form shaded areas in streets, thereby protecting pedestrians from direct sunshine [50]. Except for the construction of necessary public facilities, new large-scale developments may not increase the original building density and disaster-causing danger.

(2) For old communities located in the center of the city, pay special attention to the creation of vegetation space and the regulation of building pattern. Dotted vegetation greening could be used in old communities with limited internal open spaces, and compounds. The interior space of courtyards could be integrated into the ecological landscape, and vegetation planting in courtyards could be encouraged and subsidized. Diverse vegetation could be planted for old communities with more open spaces. Although the disaster-causing danger is significantly affected by building density, small-scale and gradual organic renewal is encouraged in old communities instead of demolishing existing buildings on a large scale. In the process of building renovation, it is suggested that terrace-backward buildings and buildings on stilts be developed, to improve the permeability of the street front and microclimate in communities.

(3) For communities in urban villages, take the lead in the establishment of rainstorm landscape and the promotion of friendly facilities for children and the elderly. Large-scale and forced relocation are not allowed in communities where low-income, vulnerable residents live. Shaded public spaces and recreational facilities should be improved in these old communities and urban villages. Permeable pavements, such as straw-inlaid bricks, could be used to pave floors to improve humidification and cooling capacity in old communities. Plants and water could be combined to set up stormwater landscapes to improve permeable areas, and rain gardens and sunken green spaces could be built to improve the diversity of the ecological landscape in the community. At the same time, it would be effective to implement heat mitigation measures and increase the health and comfort of residents.

(4) For communities on the edge of the city, lay emphasis on increasing greening layout and improving infrastructure service. Communities in the urban fringe lack adequate housing, shade, and green open space. There are few new buildings, so optimizing the layout of buildings is not an option to improve conditions in these communities. Illegal buildings and dangerous houses can be demolished, through which building density can be reduced to improve the level of openness and heat dissipation. Green patches in communities could be used to build pocket parks with trees, shrubs, and grasslands.

## 5. Conclusions

Based on 973 communities in Wuhan, this study determined the measurement and weight system of high-temperature risk assessment, constructed a risk assessment model for high-temperature disasters, and carried out a comprehensive risk assessment of high-temperature disasters. The study identified different types of high-temperature risk areas, analyzed the influence of built environmental factors and their spatial heterogeneity, and proposed optimization strategies to mitigate the risk of high-temperature disasters, providing planning guidance for the optimization of the built environment of high-risk communities.

On the whole, this study draws the following three conclusions.

(1) The comprehensive risk of high-temperature disasters in Wuhan is mainly manifested in the following four aspects. (a) The spatial distribution of high disaster-causing danger in the community is very consistent with its surface temperature. The high-risk areas are spatially concentrated and distributed along the axis, and the medium-risk areas extend to the low-risk areas. (b) The spatial distribution of disaster-generating sensitivity in the community shows the spatial characteristics of the clustered distribution of high sensitivity areas. Because of the high vegetation coverage and water proximity, there is minimal environmental sensitivity of communities along the Yangtze and Han rivers, and around East and South Lake. (c) The spatial distribution of disaster-bearing vulnerability in the community shows a trend of gradually decreasing outward with Hankou as the center, among whom the elderly and infants are highly exposed to high-temperature disasters. (d) The comprehensive risk level of high-temperature disasters in most communities is medium, with low and high value clustering in some communities. It shows a spatial pattern of multi-axial extension in high-risk areas and distribution of low-risk areas along the rivers and lakes.

(2) The main factors that influence built environment community surface temperature in different risk areas in Wuhan are different. (a) The main negative influencing factors of community surface temperature in high-risk areas are NDVI, proximity to water, and FAR. (b) The main positive influencing factor of surface temperature in low-and medium-risk areas is building density.

(3) Mitigating high-temperature disaster risk in the community should be based on three optimization strategies: (a) shaping a compact and intensive built environment to reduce disaster risk in the community; (b) creating a built environment with blue and green interweaving to reduce the probability of community disaster; and (c) creating a built environment with complete facilities and improving the community’s ability to resist disasters. A range of optimization strategies should be performed for the built environment of high-risk communities in different urban areas.

In order to mitigate high-temperature disaster risk in communities, we should carry out integrated and optimized actions on the overall situation and differentiate improvements in the local space in high-risk communities. The findings of this study extend research in the field of high-temperature disaster risk assessment and built environment impact factors; aid understanding of the development of laws and regulations, and the driving mechanisms of urban thermal environment elements; and provide a theoretical basis and decision support for urban planning, urban construction, and improvement of human living environment quality.

## Figures and Tables

**Figure 1 ijerph-19-00183-f001:**
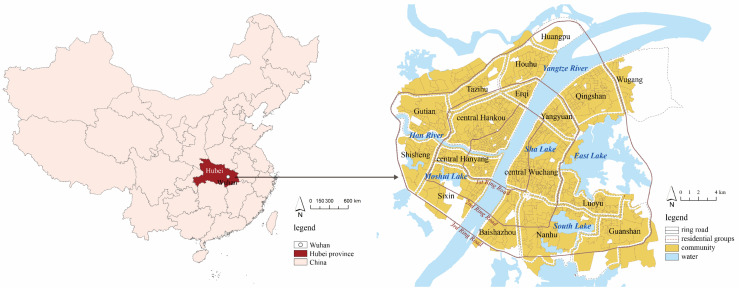
The location of downtown Wuhan.

**Figure 2 ijerph-19-00183-f002:**
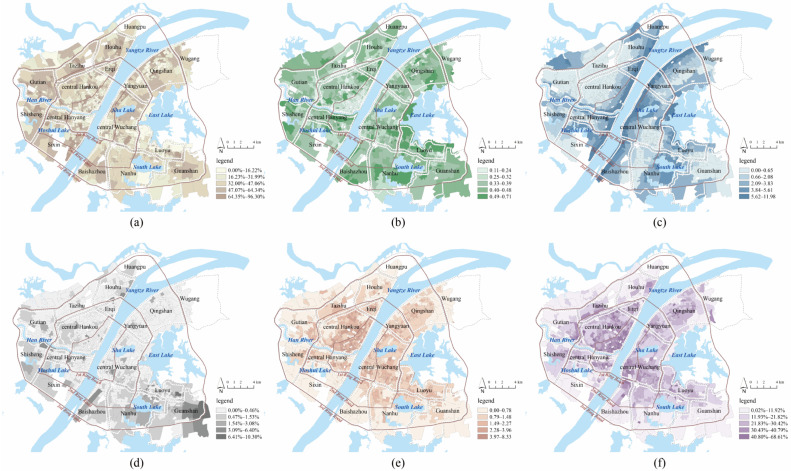
(**a**) Spatial patterns of impervious land coverage; (**b**) Spatial patterns of normalized difference vegetation index; (**c**) Spatial patterns of proximity to water; (**d**) Spatial patterns of bare land coverage; (**e**) Spatial patterns of floor area ratio; (**f**) Spatial patterns of building density.

**Figure 3 ijerph-19-00183-f003:**
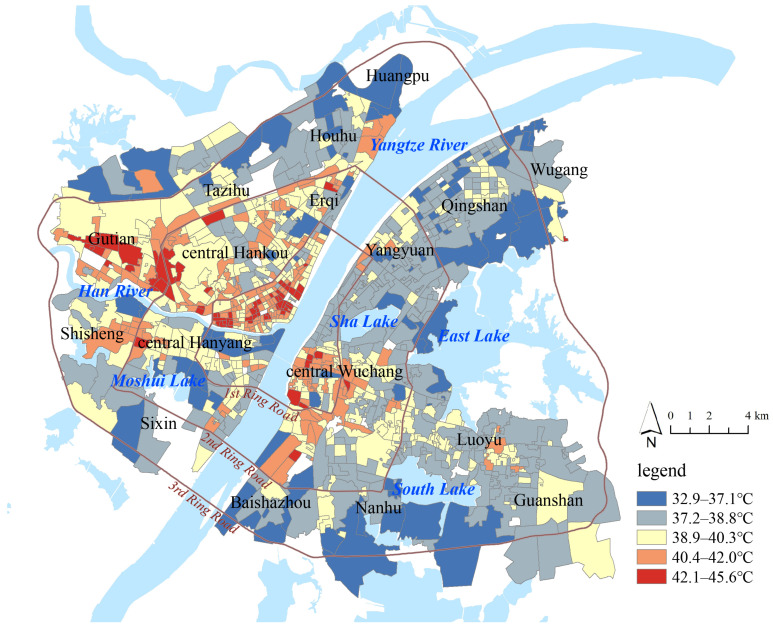
Map of community LST in downtown Wuhan.

**Figure 4 ijerph-19-00183-f004:**
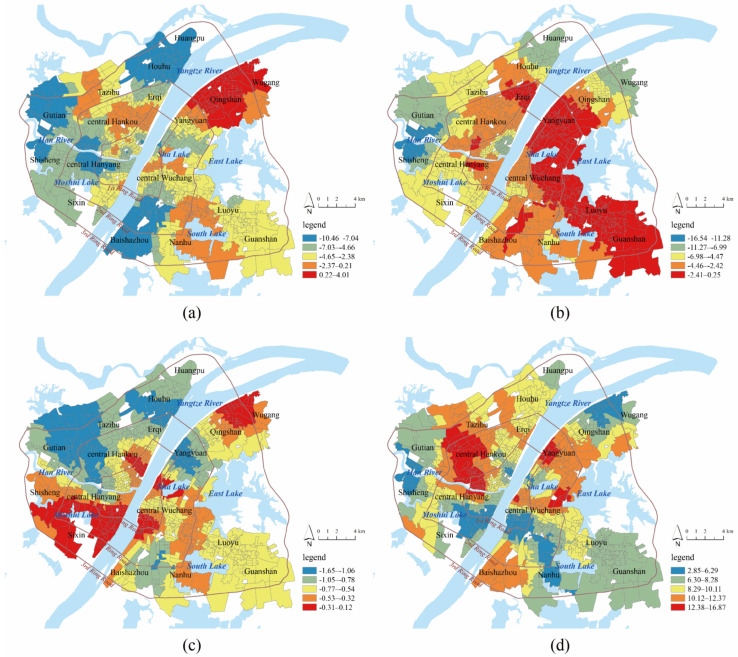
(**a**) Spatial patterns of local B coefficients of normalized difference vegetation index in GWR model; (**b**) Spatial patterns of local B coefficients of proximity to water in GWR model; (**c**) Spatial patterns of local B coefficients of floor area ratio in GWR model; (**d**) Spatial patterns of local B coefficients of building density in GWR mode.

**Figure 5 ijerph-19-00183-f005:**
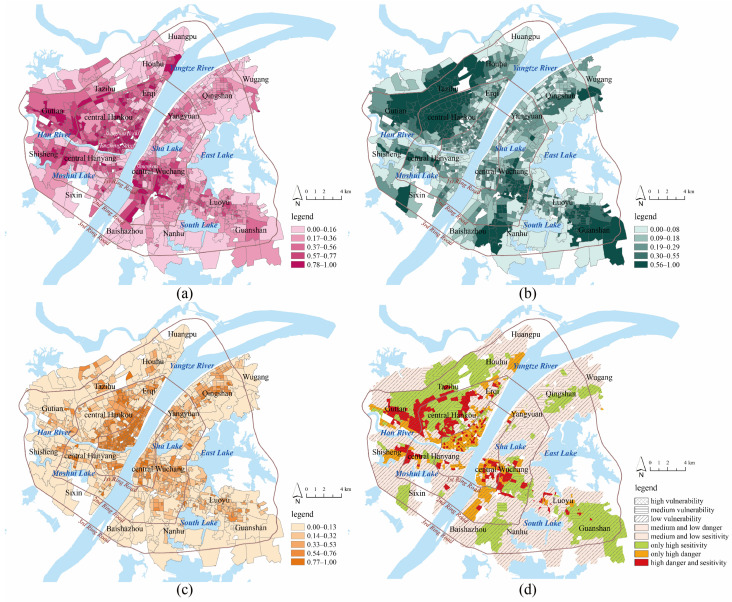
(**a**) Assessment of disaster-causing danger; (**b**) Assessment of disaster-generating sensitivity; (**c**) Assessment of disaster-bearing vulnerability; (**d**) Assessment of comprehensive risk.

**Table 1 ijerph-19-00183-t001:** Statistics of built environment factors.

Category	Metrics	Mean	Median	Min.	Max.
Land cover	Impervious land coverage	37.64%	37.50%	0.00%	96.30%
	Normalized difference vegetation index	0.34	0.34	0.11	0.71
	Proximity to water	2.80	2.75	0.00	11.98
	Bare land coverage	0.27%	0.00%	0.00%	10.30%
Development intensity	Floor area ratio	1.46	1.38	0.00	8.33
	Building density	25.49%	25.58%	0.02%	68.61%

**Table 2 ijerph-19-00183-t002:** Results of the OLS regression (adjusted R^2^ = 0.56, AICc = 2851) and multicollinearity test.

Factors	Standardized B Coefficient	Sig.	Tolerance	VIF
Impervious land coverage	−0.25	0.000	0.40	6.15
Normalized difference vegetation index	−0.28	0.000	0.72	1.38
Proximity to water	−0.30	0.000	0.79	1.25
Bare land coverage	0.033	0.061	0.91	1.11
Floor area ratio	−0.25	0.000	0.45	2.18
Building density	0.54	0.000	0.40	2.47

**Table 3 ijerph-19-00183-t003:** Results of GWR model (adjusted R^2^ = 0.75, AICc = 2467).

Metrics	Min.	Lower Quartile	Median	Upper Quartile	Max.	Mean	DIFF of Criterion	+ (%)	− (%)
Normalized difference vegetation index	−10.46	−5.82	−3.67	−2.12	4.01	−3.66	−47.29	9.66	90.34
Proximity to water	−16.54	−5.44	−3.67	−2.31	0.24	−4.19	−12.21	1.12	98.88
Floor area ratio	−1.65	−0.88	−0.62	−0.41	0.12	−0.64	−8.76	0.61	99.39
Building density	2.85	7.39	8.98	10.62	16.87	9.09	−38.32	100	0

**Table 4 ijerph-19-00183-t004:** Pivot table for disaster-causing danger, disaster-generating sensitivity and disaster-bearing vulnerability.

Disaster-causing danger	Disaster-bearing vulnerability					
		low	relatively low	medium	relatively high	high
	low	111	26	7	6	1
	relatively low	98	62	46	22	8
	medium	56	67	45	48	13
	relatively high	20	42	45	44	39
	high	19	22	27	40	59
Disaster-generating sensitivity	Disaster-bearing vulnerability					
		low	relatively low	medium	relatively high	high
	low	99	26	23	18	18
	relatively low	67	64	39	54	28
	medium	48	44	43	35	33
	relatively high	26	39	24	30	25
	high	64	46	41	23	16
Disaster-causing danger	Disaster-generating sensitivity					
		low	relatively low	medium	relatively high	high
	low	70	39	22	3	17
	relatively low	59	77	35	24	41
	medium	26	60	46	41	56
	relatively high	14	38	53	43	42
	high	15	38	47	33	34

**Table 5 ijerph-19-00183-t005:** Planning strategies.

Category	Priority Policy and Measures
Communities being built or to be built	Organize land use structure
	Optimize architectural composition
Old communities in the center of the city	Create vegetation space
	Regulate building pattern
Communities in urban villages	Create rainstorm landscape
	Establish child-friendly facilities
	Establish elderly-oriented facilities
Communities on the edge of the city	Increase greening layout
	Improve infrastructure service

## Data Availability

The data presented in this study are available on request from the corresponding author.

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
