# Peer review of "High-Temperature Disaster Risk Assessment for Urban Communities: A Case Study in Wuhan, China"

_ijerph, 2021, doi:10.3390/ijerph19010183_

Round 1

Reviewer 1 Report

This article aims to assess the risk of high-temperature disasters in urban communities providing theoretical support based on GWR to analyze the spatially different effects of influencing built environment factors.

The paper content has a relevant value in the field of investigation. A comprehensive assessment of disaster risk was developed by integrating three dimensions: disaster-causing danger, disaster-generating sensitivity, and disaster-bearing vulnerability. A series of optimization strategies of the built environment is proposed to mitigate the risk of high-temperature disasters. 
The study is comprehensive, correctly designed, and technically sound. The methodological approach is accurate and appropriate to analyze the specific issue.

The figures and tables are clear and properly show the data. I suggest improving the readability of Figures 2. The description of the planning strategies reported in the final section of the paper (lines 421-447) could be reported in a listed form suggesting a priority setting based on the impact of each intervention on the built environment to prevent and mitigate high-temperature disasters. 

Reviewer 2 Report

The work addresses "High-temperature risk disaster" in the context of the city of Wuhan, China. This study is important for the scope of the Journal, but needs few adjustments.

In the Methodology:
Lines 166-167: about "The community LST was calculated by averaging all values ​​of pixels in the extent, and pixels on building roofs were excluded.", explain why this exclusion.

Results:
Line 246: about "Moran's I value", inform in Methodology;
Table 2. cite what is VIF;
Figure 4, inform the measurement units;
Lines 280-281: check if the sentence is correct "At the same time, most permeable land is occupied by high-density buildings,...";
Lines 311-312: on "...environmental quality is quite poor with overly high 311 building density.", inform that it is poor when compared to rural areas, but better than central urban areas, No?
Line 377: Was the word "fortunately" used properly?

Policy Implications: the discussion is poor, especially without using the scientific literature. For example, you should further explore the discussion on public health, green corridors, green and blue areas.
Lines 432-433: About the sentence "...is not allowed and shaded public spaces and recreational facilities should be improved.", you can improve by being more precise.
Lines 433-434: About "certain building height could be maintained to form shaded areas on streets," how much would be the proper average height? Use scientific literature.

Conclusions:
 Lines 486-487: about "...two aspects: overall and differential optimization of high-risk communities in different urban areas.", be more precise in writing these two aspects.

Reviewer 3 Report

The article is well presented, I would suggest very few a minor adjustments:

Line 35: explain why / in which why urban high temperature disasters consume a large amount of energy

Line 41: add a reference to the "claimed 35,000 lives"

Line 418: "In three years" specify when the 3 years will start for example it could be rephrased as "From 20xx China aims to reconstruct in three years...."

References are indicated with bullet points while in the text with numbers this does not make possible to associate the number with the reference

The delimitation of the 1st, 2nd and 3rd rings are not very readable maybe use a different character  or colour

Round 2

Reviewer 2 Report

The authors carried out revisions that left the new version of the article suitable for publication.